# Course of Lower Respiratory Tract Infection in Young People Treated at the Military Hospital of Smolensk Garrison with Detected *Mycoplasma pneumoniae* Carrying a Macrolide-Resistant Mutation in 23S rRNA Gene

**DOI:** 10.3390/pathogens12010103

**Published:** 2023-01-07

**Authors:** Inna Alexandrovna Edelstein, Olga Vladimirovna Ivanova, Oleg Igorevich Romashov, Roman Sergeevich Kozlov

**Affiliations:** 1Laboratory Molecular Diagnostics, Institute of Antimicrobial Chemotherapy, Smolensk State Medical University Federal State Budgetary Educational Institution of Higher Education «Smolensk State Medical University» of the Ministry of Healthcare of the Russian Federation, Smolensk 214019, Russia; 2Pulmonology Department, Brach No 4 of FSGI «1586 MCH» of the Ministry of Defense of the Russian Federation, Smolensk 214012, Russia

**Keywords:** community-acquired pneumonia, *Mycoplasma pneumoniae*, 23S rRNA, acute bronchitis

## Abstract

We evaluated the effect of macrolide-resistant mutations in the *Mycoplasma pneumoniae* 23S rRNA gene on the severity of lower respiratory tract infections in immunocompetent young adults treated at the Smolensk Military Hospital between 25 October 2017, and 17 November 2021. All analyzed cases represented a non-severe infection of the lower respiratory tract: 44 case histories with community-acquired pneumonia and 20 cases with acute bronchitis. The presence of mutations in the gene 23S rRNA of *M. pneumoniae* was determined with standard Sanger sequencing. The macrolide-resistant genotype was found in 4/44 (9.1%) of the samples of the patients with pneumonia and in 3/20 (15%) of the samples of the patients with acute bronchitis. The analyzed cases with identified *M. pneumoniae* carrying a mutation in the 23S rRNA gene did not show any differences in the clinical presentation in terms of disease severity caused by *M. pneumoniae* with the wild-type (WT) phenotype.

## 1. Introduction

Community-acquired pneumonia (CAP) is one of the most-prevalent infectious diseases and a complex problem for the healthcare system. In the Russian Federation, the incidence of CAP in 2019 was 51.89 per 10,000 inhabitants, which is 5.5% higher than in 2018 (49.167 per 10,000). The percentage rate of lethal outcomes for CAP remains high, 1–3% in young and middle-aged adults without concomitant diseases [1,2]. Acute bronchitis (AB) is also one of the most-pressing problems of modern pulmonology, which is associated with a high incidence of 30–40% [3].

In the RF Military Forces in 2018, the incidence of CAP was 277 per 10,000 conscripts and 52 per 10,000 contract servicemen [3]. Due to the presence of long-term contacts in isolated and semi-isolated groups of military personnel, favorable conditions for the pathogen’s circulation remain, which leads to rapid and widespread infection among members of organized groups [4,5].

In the vast majority of cases of acute bronchitis in adults, the causative agents are respiratory viruses. In rarer cases, the disease is caused by *Bordetella pertussis*, *Mycoplasma pneumoniae*, and *Chlamydophila pneumoniae*, which account for no more than 5–7% of all cases [3]. Most CAP cases are associated with a relatively small range of pathogens, which include *S. pneumoniae*, *H. influenzae*, *respiratory viruses*, *Enterobacteria*, *S. aureus*, and *L. pneumophila* [3]. A significant role in the etiology of CAP is played by “atypical” pathogens: *Mycoplasma pneumoniae*, *Chlamydophila pneumoniae*, and *Legionella pneumophila*. In a non-serious course, the proportion of *M. pneumoniae* and *C. pneumoniae* in the etiological structure reaches cumulatively 20–30% [6].

*M. pneumoniae* is one of the most-common “atypical” causative agents of CAP. In the RF, according to data from various studies, *M. pneumoniae* is identified in 9.5% to 16.7% of the cases in the non-severe pneumonia group among conscripts [7,8,9,10].

As a rule, pneumonia of mycoplasmal etiology has a mild course and has a favorable outcome in young people. The severe course is characterized by fever of high severity and rapid progression of respiratory failure and can lead to death [10]. 

The most-common clinical signs of pneumonia caused by *M. pneumoniae* are: a dry cough, which can be paroxysmal and prolonged, fever (at the initial stage, usually accompanied by subfebrile temperature), and intoxication syndrome manifested by headache, general weakness, myalgia, and the presence of a runny nose and itchy throat in the first days of the disease [5]. Physical examination data in young people are more often undescriptive: there are no changes in percussion sound; upon auscultation, dry or single fine crackles may be heard; only reduced vesicular breathing may be revealed. Routine methods of examination for patients with signs of lower respiratory tract infection often do not reveal the pathological changes typical of pneumonia. After 3–7 days from disease manifestation, radiological changes appear, which do not always have a classic pattern, i.e., infiltration of the lung tissue. They may, rather, present with increased pulmonary vascularity, which makes it difficult to diagnose pneumonia [5]. Taking into consideration the non-specific nature of the symptoms of lower respiratory tract infection, laboratory diagnostics is mandatory to identify the etiology and to determine the therapeutic approach for the patient. In this case, sputum analysis has the greatest diagnostic value, due to the high concentration of the pathogen in the lower respiratory tract. However, given the unproductive cough characteristic of the clinical manifestation of pneumonia and the inability to obtain sputum for examination, specimens from the upper respiratory tract—scrapings of the posterior pharynx and nasopharynx—are usually used [11]. The culture method is the most time-consuming and expensive because *M. pneumoniae* belongs to the slow-growing bacteria, which are very demanding with respect to the conditions of cultivation. The most-accurate serological diagnosis is provided by determining IgM and IgG in paired sera collected at intervals of 2–3 weeks, and evidence of mycoplasma infection is considered to be at least a fourfold increase in antibody titer. The long testing period does not allow these methods to be used in in-patient practice. Polymerase chain reaction (PCR) is currently most important in the verification of mycoplasma infection, and the use of this technology in real-time and in a multiplex format makes it possible to simultaneously identify several etiologically significant pathogens and prescribe etiotropic therapy, as well as to adjust the results of empirical therapy [12].

Macrolides are highly effective drugs against “atypical” pathogens and are included in first-line therapeutic regimens for the treatment of community-acquired pneumonia and acute bronchitis of mycoplasma etiology. They provide a high concentration in bronchial mucus and lung tissue, have a high safety profile, and have a unique anti-inflammatory effect [13]. 

Since 2000, the prevalence of secondary resistance to macrolides in *M. pneumoniae* has been established, and the current level of resistance to these drugs varies from 0 to 15% in different countries in Europe and the USA and up to 30% in Israel, and the highest rate characteristic of Asian countries ranges from 90–100% [14,15]. The mechanisms of macrolide resistance in clinical isolates *M. pneumoniae* are now well described. One of the lead mechanisms is the occurrence of mutations in genes of the peptidyl transferase loop in the V domain of 23S rRNA in *Mycoplasma pneumoniae*, which leads to a decrease in the affinity for the drug [16]. 

In the available reports on the clinical manifestations of pneumonia caused by such pathogens, it is usually noted that there are no differences in the clinical course and severity of the disease in patients infected with macrolide-resistant strains and patients infected with macrolide-sensitive strains of *M. pneumoniae* [15,17]. Clinical pneumonia symptoms and laboratory and X-ray results, as well as prognostic data were the same, irrespective of the resistance profile of the causative agent. According to available publications, the clinical significance of the resistance to macrolides in such patients was usually limited to a prolonged duration of symptoms and did not increase the risk of complications [17,18]. 

Our aim was to assess the influence of mutations in the 23S rRNA gene of *M. pneumoniae* on the severity and outcomes of lower respiratory tract infections in immunocompetent young individuals treated at a military hospital. Given the relevance of the problem, our study focuses on the analysis of the course of community-acquired pneumonia in patients with macrolide-resistant *M. pneumoniae.*

## 2. Materials and Methods

The retrospective analysis included 155 case histories of young patients aged 18 to 44 years (*n* = 95 CAP and *n* = 60 AB) with signs of lower respiratory tract infections, treated at the Smolensk Military Hospital between 25 October 2017, and 17 November 2021. All patients were young adults aged 20 ± 1.9 years, males, without any concomitant somatic pathology. During the hospitalization, their medical history was compiled; a physical examination was conducted, as were a complete blood count, urinalysis, C-reactive protein test, blood chemistry, electrocardiography, and chest X-ray performed. The results of these tests were used to establish the diagnosis of CAP. In all hospitalized patients who did not receive macrolide antibiotics as part of their outpatient care, prior to prescribing them an antibacterial therapy, throat swabs were taken to test for the presence of *M. pneumoniae* DNA. The presence of *M. pneumoniae* DNA was determined at the local hospital laboratory by means of real-time PCR with the commercial kit AmpliSens^®^
*Mycoplasma pneumoniae*/*Chlamydophila pneumoniae*-FL (Federal Budget Institution During the follow-up of Science Central Research Institute of Epidemiology (FBIS CRIE), Moscow, Russia). Prior to PCR, DNA was isolated using the Ribo-Prep kit (FBIS CRIE). In patients with a positive PCR test result, the test was repeated at the end of treatment. The presence of macrolide-resistance-associated mutations in domain V of the 23S rRNA gene was investigated using Sanger sequencing at the laboratory of the Institute of Antimicrobial Therapy. Briefly, the DNA extracted from throat swab specimens was used to PCR amplify a 747 bp DNA fragment corresponding to the domain V region of the 23S rRNA gene using the primers Mpn23sSeqF, 5′-CGTCCCGCTTGAATGGTGTAAC-3′, and Mpn23sSeqR, 5′-GCGCTACAACTGGAGCATAAG-3′. The corresponding PCR products were purified by exonuclease I and shrimp alkaline phosphatase treatment and sequenced on both strands using the same primers and a BigDye^®^ Terminator v3.1 Cycle Sequencing Kit on the Applied Biosystems 3500 Genetic Analyzer (Life Technologies, Carlsbad, CA, USA). DNA sequences were compared to the reference sequence of the *M. pneumoniae* 23S rRNA gene (GenBank Accession No. NR_077056.1). The nucleotide positions of mutations associated with macrolide resistance were identified and reported according to the conventional *E. coli* numbering.

## 3. Results

In the primary test, *M. pneumoniae* was detected by PCR in respiratory specimens of 44/95 (46.3%) patients with CAP. On admission to the hospital, all patients with *M. pneumoniae* CAP had a clinical presentation of non-severe CAP, an SMRT-CO score of 0 points, with the prevalent complaints being general weakness, a cough with scant sputum, a fever of 37.5 °C to 39.5 °C, chills, and an itchy throat. Respiratory failure, hemoptysis, and pain in the chest were not reported, nor pneumonia-related complications. In the vast majority of cases, the manifestation of a systemic inflammatory response in patients was insignificant; leukocytosis of up to 10 × 10^9^/L without left shift was detected in the complete blood count. In one case, leukocytosis of 15 × 10^9^/L with an 8% increase in the number of neutrophils was reported upon admission. Blood chemistry parameters and urinalysis were normal. Because all patients had a non-productive cough on admission to the hospital, no general clinical and microbiological sputum analysis was performed.

In 42 patients with *M. pneumoniae* CAP, the X-ray showed unilateral lung infiltrate in no more than two lung segments, and in two patients, both lung lobes were affected, with infiltrate present in three segments of the right lung and one segment of the left lung, which, however, did not cause a more severe condition of the patient nor clinical manifestations of pneumonia. None of the investigated cases showed radiological signs of lung tissue consolidation and destruction or pleuritis. A typical macrolide-resistance-associated mutation, A2058G, in the domain V region of the 23S rRNA gene, was detected in the samples of 4/44 (9.1%) patients with *M. pneumoniae* CAP, 3 of whom were members of the same military unit. The remaining patients revealed the wild-type (macrolide-susceptible) genotype of *M. pneumoniae*.

In 60 patients with signs of lower respiratory tract infections (cough, subfebrile body temperature, non-expressed intoxication syndrome, dry rales in lung auscultation), laboratory and radiological signs of pneumonia were not reported. Due to the lack of other alternative reasons, the condition was assessed as AB. *M. pneumoniae* was found in 20/60 (33.3%) patients with AB. The macrolide-resistant A2058G genotype was found in 3/20 (15%) of cases. All analyzed cases of *M. pneumoniae* AB were a non-severe infection; there were no complications and prolonged inflammatory processes. Antibacterial therapy with azithromycin or clarithromycin in standard dosages was administered in all cases of *M. pneumoniae* AB regardless of the macrolide resistance status, which was determined retrospectively. At the end of the therapeutic course, all 17 patients with the macrolide-susceptible genotype in the primary test were *M. pneumoniae*-negative in the follow-up test; however, two of the three patients initially with the macrolide-resistant genotype revealed the presence of *M. pneumoniae* DNA in the follow-up PCR test, despite the complete resolution of respiratory symptoms. 

The CAP cases were subject to a more detailed analysis (Figure 1). 

According to the CAP treatment guidelines [8], empirical antibacterial therapy with parenteral third-generation cephalosporin (ceftriaxone, 2 g/day) was started in the hospital for all patients with CAP before the PCR test result for *M. pneumoniae* was obtained. When *M. pneumoniae* was detected, oral azithromycin or clarithromycin in standard dosage was added. Three patients with macrolide-resistant *M. pneumoniae* received azithromycin 500 mg/day, and one patient received clarithromycin 500 mg/BID. In the latter patient, the combination therapy with ceftriaxone and clarithromycin was started already at the initial stage, because of the prior epidemiological evidence of *M. pneumoniae* infection in other patients of the same military unit. In all examined patients with macrolide-susceptible and macrolide-resistant *M. pneumoniae*, clinical improvement was reported and the fever was resolved two days after the macrolides were administered. No complications were observed, and no follow-up adjustment of the antibacterial therapy was necessary. Likewise, in all patients with *M. pneumoniae* CAP, clinical, laboratory, and radiological resolution of pneumonia occurred in 10–16 days, regardless of the macrolide resistance status. No residual changes, such as adhesions or pneumofibrosis, were reported. Complete blood count and blood chemistry were normal. No changes in the ECG and no post-infection asthenia were seen. However, in the follow-up throat swab PCR test, *M. pneumoniae* DNA was detected in one of the four patients with initially macrolide-resistant *M. pneumoniae*, but none of the 40 patients with initially susceptible *M. pneumoniae*. Since the only patient with a positive follow-up PCR test had no clinical manifestations of infection, the antibacterial therapy was discontinued and no new antibiotics were administered. All patients were discharged in satisfactory condition.

We provide here the description of a typical clinical case of CAP due to *M. pneumoniae* with an A2058G mutation in the 23S rRNA gene.

A 21-year-old male patient P. was hospitalized on 27 November 2017, in the Pulmonology Department of the military hospital. On admission, his complaints were general weakness, a predominantly dry cough, an itchy throat, and a fever of up to 39.5 °C.

*Disease history*: The patient had been sick with a dry cough and general weakness for three days after an episode of exposure to cold weather. He sought medical care on 27 November 2017, when the dry cough increased, the body temperature increased to 39.5 °C, and a headache appeared. He did not take any medication. Upon examination at the out-patient department of the military hospital, a chest X-ray was performed; infiltration was found in the sixth segment of the left lung, and the patient was referred to the hospital for inpatient treatment. The epidemiological data indicated that there were recent cases of respiratory infection with *M. pneumoniae* among members of the same military unit.

*On admission*, the patient was fully conscious. His body temperature was 38.8 °C. The condition was satisfactory. There were no tension symptoms. The skin was of normal color, clean, and without cyanosis; the SpO2 was 98% at room air. Peripheral lymph nodes were not swollen on palpation; visible mucosa was pink and clear; the pharynx was not hyperemic. The tongue was clear and moist. The percussion lung sound was clear. On auscultation, breathing was vesicular; on the left side, the breathing was slightly decreased at the lower regions. There were no abnormal sounds. The respiratory rate was 17 breaths per minute. Cardiac sounds were clear and rhythmic. Tachycardia of the heart rate of 120 beats per minute was reported. The blood pressure was 120/70 mmHg. The abdomen was soft and painless upon palpation. The inferior border of the liver was non-palpable. There was no visible edema. Bowel and bladder functions were normal.

*Laboratory tests* were performed on 28 November 2017. Complete blood count showed: WBC, 7.0 × 10^9^/L; RBC, 5.02 × 10^12^/L; hemoglobin, 149 g/L; neutrophils, 1%; segmented neutrophils, 65%; monocytes, 10%; eosinophils, 1%; lymphocytes, 23%; platelets 207 × 10^9^/L; ESR, 17 mm/h. No abnormalities were revealed in the urinalysis and blood chemistry. In view of the fact that the patient had a non-productive cough, no sputum analysis was conducted, and for the detection of causative agent, a throat swab sample was taken. 

*Chest X-ray in anteroposterior and lateral views*, dated 27 November 2017, revealed a low-intensity infiltration in the S6 projection of the left lung. The roots were structural. The pleural recesses were clear. The cardiac shadow was unchanged (Figure 2).

In *electrocardiography* dated 28 November 2017, the cardiac rhythm was sinus with a heart rate of 57–79 bpm. A sinus arrhythmia was noted. The cardiac axis was vertical.

*The clinical diagnosis* was defined as a non-severe community-acquired focal pneumonia with localization in S6 of the left lung, without respiratory failure, with a SMRT-CO of 0 points.

As the starting antibacterial therapy, ceftriaxone 2 g/day was administered. Oral detoxication, mucolytic therapy, and diclofenac and paracetamol as antipyretics were administered according to the indications. Upon receipt of positive PCR test results for *M. pneumoniae* on 29 November 2017, azithromycin 500 mg/day was added to the treatment. The body temperature returned to normal, and the headache resolved two days after azithromycin treatment was initiated. On the fourth day, the cough subsided and became productive with an insignificant quantity of light sputum. On 1 December 2017, sequencing of the *M. pneumoniae* 23S rRNA gene was performed, which revealed a macrolide-resistance mutation, A2058G. Taking into account the improvement of the patient’s condition and the resolution of the respiratory symptoms, the antibacterial therapy was not changed. The patient received physical therapy from 4 December 2017.

On 7 December 2017, the follow-up *complete blood count* showed: WBC, 8.4 × 10^9^/L; RBC, 5.1 × 10^12^/L; hemoglobin, 153 g/L; neutrophils, 2%; segmented neutrophils, 51%; monocytes, 10%; eosinophils, 1%; lymphocytes, 36%; platelets 380 × 10^9^/L, ESR, 10 mm/h. Urinalysis, blood chemistry, and ECG revealed no abnormalities. *Chest X-ray in anteroposterior and lateral views* performed on the same day showed the resolution of infiltration with slightly increased local vascularity and consolidation.

The follow-up PCR test for *M. pneumoniae* performed before the patient’s discharge from the hospital was negative. The patient was discharged in satisfactory condition on the 15th day. 

All studied cases of *M. pneumoniae* CAP were considered as non-severe based on clinical, radiological, and laboratory data. The pulmonary involvement was mild; dry cough was the predominant respiratory symptom; intoxication syndrome was not evident, which is typical of *M. pneumoniae* CAP. In all cases, fever and intoxication syndrome were relieved about 48 h after administration of macrolides (azithromycin or clarithromycin) in standard dosages. No respiratory support nor prolonged antibacterial therapy were required. Most notably, no difference was observed between the patients infected with macrolide-resistant and macrolide-susceptible *M. pneumoniae* in terms of clinical symptoms, severity of pneumonia, laboratory results, radiographic findings, and clinical outcomes of treatment with macrolides. Replacement of macrolides with other antibiotics active against *M. pneumoniae*, e.g., fluoroquinolones, was not required either, regardless of the macrolide resistance status of *M. pneumoniae*.

It should be noted that the use of PCR for the detection of “atypical” pathogens allowed for rapid correction of antibacterial therapy, as most patients had received the starting therapy with a third-generation cephalosporin, and macrolides were added after obtaining a positive PCR result for *M. pneumoniae*. In a few cases, however, where prior epidemiological evidence suggested *M. pneumoniae* infection in members of the same military unit, patients received a combination of cephalosporin and macrolide already as the starting therapy. Therefore, timely examination of patients for “atypical” pathogens and communicating epidemiological findings were crucially important both for successful treatment and for antibiotic stewardship. Moreover, this helped limit the further, potentially latent, spread of the pathogen in a closed military community.

In our study, *M. pneumoniae* was found as a causative agent of non-severe lower respiratory tract infection (AB and CAP) in 64/155 (41.3%) patients in the Smolensk military garrison. This prevalence is relatively high as compared to other similar reports from Russia [7,8,9,10] and may be due to the local outbreak of *M. pneumoniae* during the study period. The macrolide-resistance genotype was detected in 7/64 (10.9%) *M. pneumoniae*-positive samples collected at patients’ admission to the hospital. In most cases, the result of DNA sequencing analysis for macrolide resistance was available retrospectively (at least two days after the primary PCR test result), already when the macrolide treatment was started. Therefore, the decision to continue macrolides did not take into account the data on the macrolide resistance status, but rather, was based on the assessment of the dynamics of the patient’s clinical condition and response to the treatment. While we did not find a difference in the clinical cure rates of patients with AB and CAP due to macrolide-susceptible and -resistant *M. pneumoniae*, we observed the difference in “microbiological” cure rates, as was assessed by follow-up PCR at patient discharge from the hospital. A positive follow-up PCR test result for *M. pneumoniae* was obtained in none of the 57 patients with macrolide-susceptible *M. pneumoniae*, but in 3 of 7 patients with macrolide-resistant *M. pneumoniae*. It should be noted, however, that the small number of cases of macrolide-resistant *M. pneumoniae* infection analyzed in this study represents a major limitation to the present research. All cases were a non-severe infection in young, otherwise healthy men, which is why our findings regarding the outcomes of macrolide treatment cannot be generalized to a wider patient population. Macrolide-resistant *M. pneumoniae* have been more often reported in children because *M. pneumoniae* infection is more frequent in this population. Nevertheless, infections in adults have also been evaluated in several studies [14]. Results from these studies suggest that macrolide resistance can be associated with prolonged symptoms of the disease and may require a change of antibiotic prescription [8,18]. Failure to eradicate *M. pneumoniae* may lead to its long-lasting persistence and may be considered a risk factor for relapse of infection. However, the association between the persistence of *M. pneumoniae* DNA positivity after the clinical cure and the risk of relapse of infection has been less well studied [19,20]. The design of our study did not allow us to assess the long-term outcomes of patients with a positive follow-up PCR test for macrolide-resistant *M. pneumoniae*, which is another limitation to the present research. Repeated PCR testing may be a useful strategy to detect *M. pneumoniae* persistence and to track its potential transmission in military units.

## 4. Conclusions

Regarding clinical presentation, no difference was observed between patients infected by macrolide-resistant and macrolide-sensitive *M. pneumoniae*. Clinical symptoms, pneumonia severity, laboratory results, radiographic findings, and prognostic factors were similar regardless of the *M. pneumoniae* susceptibility to macrolides. The widespread dissemination of *M. pneumoniae* in patients with CAP 44/95 (46.3%) and the detection of macrolide-resistant *M. pneumoniae* (9.1%) indicate the need for timely laboratory diagnosis and selection of adequate antibiotic therapy. 

## Figures and Tables

**Figure 1 pathogens-12-00103-f001:**
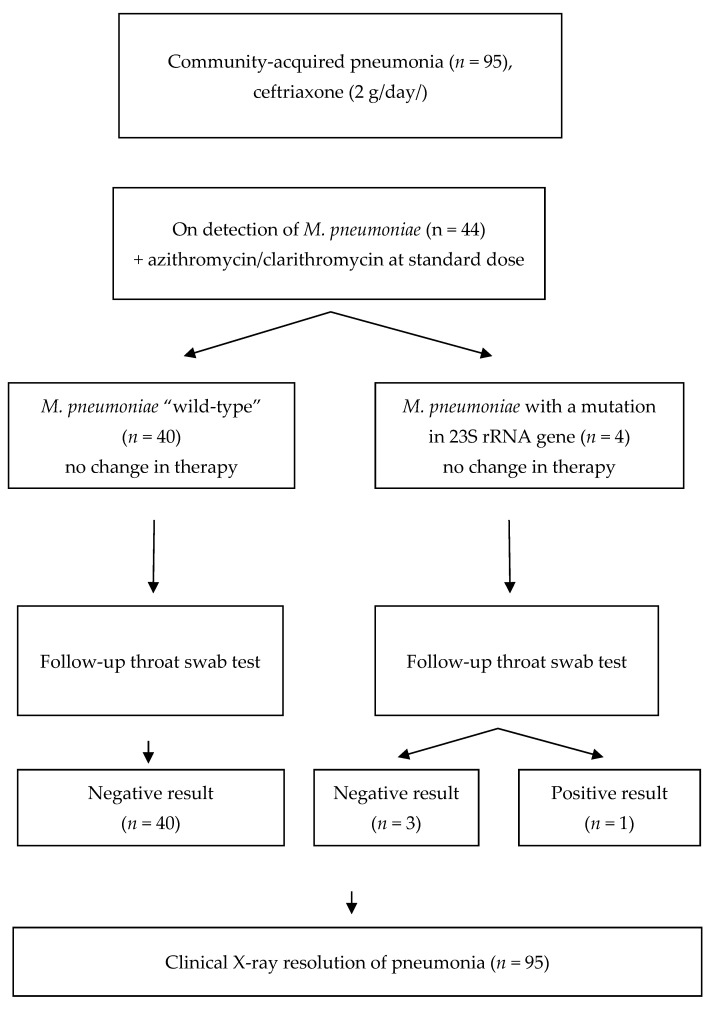
Management of the patients with CAP enrolled in the study.

**Figure 2 pathogens-12-00103-f002:**
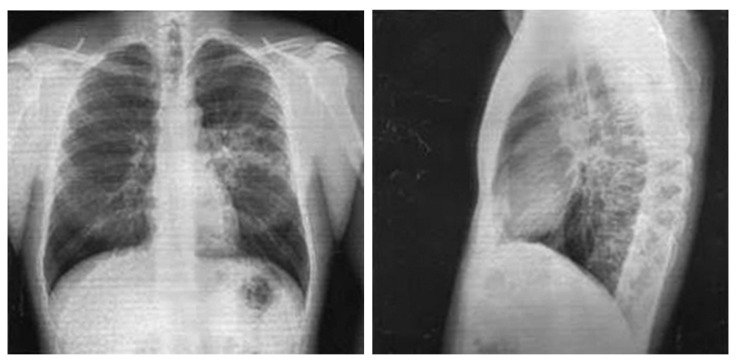
Chest X-ray of Patient P. (28 November 2017).

## Data Availability

Not applicable.

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
