# Peer review of "Course of Lower Respiratory Tract Infection in Young People Treated at the Military Hospital of Smolensk Garrison with Detected Mycoplasma pneumoniae Carrying a Macrolide-Resistant Mutation in 23S rRNA Gene"

_pathogens, 2023, doi:10.3390/pathogens12010103_

Round 1

Reviewer 1 Report

This is a very interesting article on Mycoplasma pneumoniae infection.

In this article the authors evaluated the impact of macrolide-resistant mutations in the 23S rRNA gene of Mycoplasma pneumoniae on the severity of lower respiratory tract infections in immunocompetent young patients treated at a military hospital in Smolensk between October 25, 2017 and November 17, 2021. The authors investigated that all cases evaluated showed a mild infection of the lower respiratory tract, including 44 cases of community-acquired pneumonia and 20 cases of acute bronchitis. The existence of mutations in the M. pneumoniae 23S rRNA gene was determined using conventional Sanger sequencing.The authors also stated that the macrolide-resistant genotype was identified in 4 (9.1%) of 44 individuals with pneumonia and 3 (15%) of 20 patients with acute bronchitis. The examined cases with detected M. pneumoniae bearing a mutation in the 23S rRNA gene did not vary from those produced by M. pneumoniae with WT (wild type) phenotype in terms of illness severity. The paper is well organized and written, however, there are few suggestions:

Minor concerns:

  1. The authors should increase the quality/size of figure 2 for better visualization.
  2. The authors should include detailed methodology in the method section.

Author Response

Dear Reviewer,
We are very grateful for your valuable comments to our manuscript. We revised the manuscript in accordance with your suggestions, and once again proofread it to minimize typographical and grammatical errors. The following are point-by-point responses to your comments.
1.    We have increased the size and improved the quality of the image in Figure 2 for better visualization.
2.    We have expanded the description of the methodology of analysis. The changes are highlighted in yellow color in the ‘Materials and methods’ section.

Reviewer 2 Report

You reported that they researched the patients with M. pneumoniae infections at Military Hospital and there was not any difference between the patients due to Macrolide susceptible M. pneumonia and those due to Macrolide resistant  ones.

This theme is very interesting. However, there are not a little report like your article, and your field is so small and particular. 

Therefore, you have to analyze more cases and the other people than army.

Author Response

Dear Reviewer,

We are very grateful for your valuable comments to our manuscript. We revised the manuscript in accordance with your suggestions, and once again proofread it to minimize typographical and grammatical errors. The following are point-by-point responses to your comments.
1.    We analyzed and described similar cases in other patient populations.
2.    We will insert the analysis of these cases in the "discussion" section.

Reviewer 3 Report

This manuscript describes a retrospective study that examines the prevalence of M. pneumoniae CAP in young patients treated in a Russian military hospital. The study focuses on examining the frequency of a mutation that causes macrolide resistance, as well as whether there is a significant difference in severity of disease as compared to M. pneumoniae without this mutation. Overall, the study shows that about 9% of the patients with CAP due to M. pneumoniae carry an isolate with the mutation. Furthermore, there is no significant difference in the severity or outcome of disease. Interestingly, in this study that after diagnosis of M. pneumoniae infection, there was an addition of azithromycin/clarithromycin treatment, supporting the early diagnosis of infection.

This is a very interesting paper, and it includes a case report of a patient with macrolide resistant mycoplasma infection. This inclusion provided some perspective of the disease. In addition, a patient, who was treated and recovered from disease, was found to still harbor the mutant mycoplasma. This suggests that there is the potential of spread of infection despite treatment.

One minor issue was that line 123 there was 44 out of 66 patients was equal to 46.3%. So there needs to be some clarification.

One limitation of this study is that all patients were male 18-44 years old, and there is a potential that there is the potential that there may be differences in females or other age groups. These topics should be included in the discussion.

Author Response

Dear Reviewer,
We are very grateful for your valuable comments to our manuscript. We revised the manuscript in accordance with your suggestions, and once again proofread it to minimize typographical and grammatical errors. The following are point-by-point responses to your comments.

1.    M. pneumoniae was found in 64/155 (41.29%) patients, of whom 44/95 (46.3%) patients with pneumonia.
2.    The age and sex of the subjects are limited by the peculiarities of military service selection. The aim of our study was precisely this sample of patients, which is reflected in the title of the article. In order to discuss the described phenomenon in other age groups and women, we added a "discussion" section.